# Kraus Constrained Sequence Learning For Quantum Trajectories from Continuous Measurement

## Abstract

Real-time reconstruction of conditional quantum states from continuous measurement records is a fundamental requirement for quantum feedback control, yet standard stochastic master equation (SME) solvers are computationally expensive and sensitive to parameter mismatch. While neural sequence models can fit these stochastic dynamics, the unconstrained predictors can violate physicality such as positivity or trace constraints, leading to unstable rollouts and unphysical estimates. We propose a *Kraus-structured* output layer that converts the hidden representation of a generic sequence backbone into a completely positive trace preserving (CPTP) quantum operation, yielding physically valid state updates by construction. We instantiate this layer across diverse backbones, RNN, GRU, LSTM, TCN, ESN and Mamba; including Neural ODE as a comparative baseline, on stochastic trajectories characterized by parameter drift. Our evaluation reveals distinct trade-offs between gating mechanisms, linear recurrence, and global attention. Across all models, Kraus-LSTM achieves the strongest results, improving state estimation quality by 7% over its unconstrained counterpart while guaranteeing physically valid predictions in non-stationary regimes.

## 1 Introduction

Continuous quantum measurement induces *stochastic* evolution of a system's state conditioned on the measurement record, producing *quantum trajectories* (Wiseman & Milburn, 2010; Jacobs, 2006; Rouchon & Ralph, 2022). Accurate trajectory inference underpins real-time state estimation and feedback stabilization in platforms such as superconducting qubits and quantum optics (Flurin et al., 2020). While physics-derived filters (e.g., SMEs) provide principled updates, they require correct models of Hamiltonians, decoherence, and measurement operators, which may be partially unknown or drifting in practice.

Data-driven sequence models offer a complementary route: given a measurement record and optional controls, learn to predict the conditional state evolution. However, naive regression on density matrices or Bloch vectors can break fundamental constraints (e.g., negative eigenvalues, non-unit trace), causing error accumulation and unphysical rollouts. This motivates *structure-preserving* learned updates that enforce quantum constraints during learning and inference.

**Key idea.** Any physically valid discrete-time quantum operation can be expressed (or approximated) as a Kraus map (Kraus, 1983; Nielsen & Chuang, 2010):

$$\mathcal{E}(\mathsf{h}) = \sum_{i=1}^{r} K_i \mathsf{h} K_i^{\dagger}, \qquad \sum_{i=1}^{r} K_i^{\dagger} K_i = \mathbb{I}, \tag{1}$$

which is completely positive and trace preserving (CPTP). In textbook continuous-measurement theory, the *physical* measurement operator for a particular outcome is typically trace-decreasing and the conditional state is obtained by renormalization (Wiseman & Milburn, 2010; Rouchon & Ralph, 2022). In contrast, we do *not* interpret our learned $\{K_i\}$ as literal measurement operators; instead, conditioned on the measurement record, the network directly learns a CPTP map that approximates the *normalized* conditional transition between density matrices. We leverage this representation as a *drop-in output layer* for modern sequence backbones.

**Contributions.**

- We introduce a *Kraus-structured layer* that maps backbone hidden states to completely positive (CP) quantum updates, yielding physically valid trajectory rollouts by construction.
- We provide a controlled *with/without Kraus benchmark* across major sequence-model families: gated recurrence (GRU/LSTM), simple recurrence (RNN), convolution (TCN), reservoir computing (ESN), and selective state-space models (Mamba).
- We evaluate accuracy and *physicality* under stochastic regime switching, highlighting which inductive biases best support rapid re-identification after dynamics change.

## 2 BACKGROUND

### 2.1 QUANTUM TRAJECTORIES AND STOCHASTIC MASTER EQUATIONS

Under continuous (weak) measurement, the conditional state $h_t$ follows a stochastic evolution. In diffusive (homodyne-like) monitoring, a common form is the stochastic master equation (SME) (Jacobs, 2006; Rouchon & Ralph, 2022):

$$d\mathrm{h} = \mathcal{L}(\mathrm{h})\, dt + \sqrt{\eta}\, \mathcal{H}[O](\mathrm{h})\, dW_t, \tag{2}$$

where $\mathcal{L}$ is a Lindbladian drift (Hamiltonian + decoherence), $\mathcal{H}$ is a measurement backaction superoperator for observable $O$, $\eta$ is detection efficiency, and $dW_t$ is a Wiener increment. Discrete-time simulation produces trajectory datasets: sequences of measurement increments (or currents) and corresponding conditional states.

### 2.2 KRAUS MAPS AND PHYSICALITY

A density matrix must satisfy $\mathrm{h} \succeq 0$ and $\mathrm{Tr}(\mathrm{h}) = 1$. A CPTP map guarantees these constraints are preserved (Kraus, 1983; Nielsen & Chuang, 2010). Several parametrizations exist for enforcing CPTP structure, including Choi-based constraints and geometric/Stiefel approaches (Ahmed et al., 2023; Ateeq et al., 2025). We use a Kraus parametrization to ensure CP (and, when desired, TP) by construction.

## 3 RELATED WORK

**Learning quantum trajectories.** Neural networks have been used to infer conditional dynamics from measurement records in superconducting qubits and related settings (Flurin et al., 2020). More broadly, machine learning has been surveyed for quantum estimation and control (Ma et al., 2025). Our work focuses on *structure* in the learned update to preserve physical constraints during rollouts.

**Learning quantum channels with physical constraints.** Kraus-operator learning has been used in quantum process tomography to ensure complete positivity and incorporate trace constraints (Ahmed et al., 2023). Recent work studies geometric parametrizations of Kraus operators for efficient optimization over CPTP maps (Ateeq et al., 2025). We adapt these ideas to *trajectory learning*, where the channel may depend on the measurement record and latent state.

**Sequence backbones for dynamical systems.** We benchmark a diverse set of backbones: GRU (Cho et al., 2014), LSTM (Hochreiter & Schmidhuber, 1997), Transformer (Vaswani et al., 2017), structured state-space models (Mamba) (Gu & Dao, 2024), and continuous-time Neural ODE models (Chen et al., 2018). Temporal convolutional networks (TCNs) are a strong non-recurrent baseline for sequence modelling (Bai et al., 2018) and are included in our main benchmark.

## 4 PROBLEM SETUP

We consider discrete-time sequences indexed by $t = 0, \ldots, T - 1$:

- Measurement record features $y_{0:t}$ (e.g., integrated homodyne current, innovations, or binned outcomes),

- Optional control inputs $u_{0:t}$ (e.g., drive amplitudes),
- Target conditional quantum states $h_{0:t}$ (density matrices or Bloch vectors mapped back to h).

Given history $(y_{0:t}, u_{0:t})$ (and optionally $\rho_0$), the model predicts $\rho_t$ at each timestep along sequences of length $T$. We evaluate three key aspects: (i) *one-step accuracy* via per-timestep fidelity, (ii) *multi-step stability* via long-horizon filtering performance on length-$T$ trajectories rather than separate $h$-step open-loop rollouts, and (iii) *physicality* through PSD/trace violations and Kraus-completeness error $\|\sum_i K_i^\dagger K_i - \mathbb{I}\|_F$ for Kraus heads.

## 5 METHOD: A KRAUS-STRUCTURED LAYER FOR GENERIC BACKBONES

Let $f_\theta$ be any causal sequence model (GRU/LSTM/Transformer/Mamba/Neural ODE) producing a hidden representation $h_t = f_\theta(y_{0:t}, u_{0:t})$. A naive predictor outputs $\widehat{h}_{t+1} = g_\theta(h_t)$, which may violate $\widehat{h}_{t+1} \succeq 0$ or $\mathrm{Tr}(\widehat{h}_{t+1}) = 1$.

### 5.1 KRAUS-STRUCTURED UPDATE

We instead predict a quantum operation and apply it to the previous state:

$$\widehat{h}_{t+1} = \Phi_\theta(h_t, \widehat{h}_t), \qquad \Phi_\theta(h, \mathsf{h}) = \frac{\sum_{i=1}^r K_i(h)\,\mathsf{h}\,K_i(h)^\dagger}{\mathrm{Tr}[\sum_{i=1}^r K_i(h)\,\mathsf{h}\,K_i(h)^\dagger]}, \tag{3}$$

Although our Kraus parametrization enforces $\sum_i K_i(\mathsf{h})^\dagger K_i(\mathsf{h}) = I$ (hence the map is CPTP up to floating-point precision), we additionally apply two numerical stabilizers: (i) Hermiticity projection $\hat{\rho} \leftarrow (\hat{\rho} + \hat{\rho}^\dagger)/2$, and (ii) trace renormalization $\hat{\rho} \leftarrow \hat{\rho}/\mathrm{Tr}(\hat{\rho})$ to cancel numerical drift accumulated over long horizons (here $T = 2000$); in exact arithmetic these are no-ops for a CPTP map. We construct $K_i(h)$ such that the induced operation is CPTP within numerical precision while remaining differentiable for end-to-end learningCattabriga et al. (2021).

### 5.2 PARAMETERIZING KRAUS OPERATORS

We map $h_t$ to an unconstrained complex matrix $V(h_t) \in \mathbb{C}^{(rd)\times d}$ and project it onto the Stiefel manifold via a thin QR decompositionAteeq et al. (2025):

$$V(h_t) = Q(h_t)R(h_t), \qquad Q(h_t)^\dagger Q(h_t) = \mathbb{I}_d. \tag{4}$$

We then reshape $Q(h_t)$ into $r$ blocks of size $d \times d$ to obtain Kraus operators $\{K_i(h_t)\}_{i=1}^r$. This yields

$$\sum_{i=1}^r K_i(h_t)^\dagger K_i(h_t) = \mathbb{I}_d, \tag{5}$$

defining a CPTP operation in exact arithmetic (and to numerical precision in floating-point). In our single-qubit experiments ($d = 2$) we use $r = 2$, so $V(h_t) \in \mathbb{C}^{4\times 2}$.

**With/without Kraus ablation.** For each backbone, we compare:

- **Unconstrained head**: direct state regression $\widehat{h}_{t+1} = g_\theta(h_t)$ with explicit trace normalization.
- **Kraus head**: structured update equation 3 with Kraus operators obtained by the Stiefel/QR projection equation 4.

### 5.3 LOSSES AND PHYSICALITY METRICS

We train with a combination of prediction and regularization losses:

$$\mathcal{L} = \underbrace{\mathbb{E}\big[\ell_{\mathrm{pred}}(\widehat{h}_{t+1}, h_{t+1})\big]}_{\text{accuracy}} + \lambda_{\mathrm{phys}} \underbrace{\mathbb{E}\big[\ell_{\mathrm{phys}}(\widehat{h}_{t+1})\big]}_{\text{optional physicality penalty}}, \tag{6}$$

where $\ell_{\mathrm{pred}}$ can be Frobenius error on h or MSE on Bloch vectors; and $\ell_{\mathrm{phys}}$ can penalize negative eigenvalues or trace deviation for unconstrained models. In all experiments in this paper, we set $\lambda_{\mathrm{phys}} = 0$ to isolate the effect of the Kraus-structured layer as an architectural constraint. At evaluation we report:

- Trace error: $|\operatorname{Tr}(\widehat{\mathsf{h}}) - 1|$,
- Positivity violation: $\max(0, -\lambda_{\min}(\widehat{\mathsf{h}}))$,
- State distance: trace distance or infidelity where applicable (Appendix C).

## 6 MODELS

We benchmark backbones spanning distinct inductive biases for stochastic filtering. All models are evaluated in two variants: (i) a baseline unconstrained predictor and (ii) a Kraus-constrained predictor that rolls out a physical state update. Detailed architectures and hyperparameters are provided in Appendix A.

**Gated recurrence (GRU/LSTM).** GRU (Cho et al., 2014) and LSTM (Hochreiter & Schmidhuber, 1997) implement causal filtering via an evolving hidden state with gates that enable selective reset/forgetting. These gates are well-matched to measurement back-action "kicks" and regime switchesMa et al. (2025).

**Simple recurrence (Vanilla RNN).** A minimal recurrent baseline without gates. It tests whether recurrence alone suffices, isolating the role of gating under Kraus-constrained rollouts.

**Reservoir computing (ESN).** A fixed random recurrent reservoir with a trained readout (Jaeger, 2001). This evaluates whether generic high-dimensional temporal features are sufficient when paired with a physical Kraus update.

**Convolutional filtering (TCN).** Causal dilated convolutions (Bai et al., 2018) provide a finite-window alternative to recurrence. We evaluate both standard and gated TCNs to test whether limited receptive-field memory can support filtering under switching.

**Selective state-space (Mamba).** Mamba (Gu & Dao, 2024)represents modern state-space sequence modeling with input-dependent selection and linear-time recurrence. It provides a strong long-context baseline with a different inductive bias from gated recurrenceGu & Dao (2024).

**Continuous-time dynamics (Neural ODE).** Neural ODEs learn continuous-time evolution via differential equations, providing a natural inductive bias for physical dynamics and serving as a strong baseline for comparison against discrete recurrent updatesChen et al. (2018).

**Supplementary baselines.** Physics-based SME filters and self-attention variants (Vaswani et al., 2017)are included in the supplementary material for completeness (Appendix D and E).

## 7 SYNTHETIC DATASET: SWITCHING CONTINUOUS-MEASUREMENT TRAJECTORIES

We evaluate on a synthetic dataset of single-qubit continuous-measurement trajectories designed to stress *non-stationary filtering*Breuer & Petruccione (2002). Each trajectory provides (i) a normalized homodyne measurement record and (ii) the corresponding ground-truth conditional density matrices. Dataset parameters and splits are summarized in Table 1.

**Orthogonal switching protocol.** To prevent trivial periodic memorization, the Hamiltonian undergoes an *orthogonal* switch (rotation axis changes from $\sigma_x$ to $\sigma_y$) at a random time $\tau$. This forces the model to perform in-context system identification from changing measurement statistics, and serves as a controlled benchmark for adaptation speed after a regime change.

Table 1: Dataset Specification for Non-Stationary Switching Dynamics.

| Item | Symbol | Value |
|---|---|---|
| System dimension | $d$ | 2 (Single Qubit) |
| Time step | $\Delta t$ | $0.005\,\text{s}$ |
| Sequence length | $T$ | 2000 steps (10 s total) |
| # trajectories | train/test | 2000/300 |
| Measurement efficiency | $\eta$ | 1.0 (Ideal) |
| Measured observable | $L$ | $\sigma_z$ |
| Measurement strength | $\gamma$ | $\mathcal{U}[0.3, 0.8]$ |
| Rabi frequency | $\omega$ | $\mathcal{U}[0.5, 4.0]$ |
| Switching Protocol | - | Orthogonal ($H \propto \sigma_x \to \sigma_y$) |
| Switching Time | $\tau$ | $\mathcal{U}[400, 1600]$ |

**Parameter randomization.** Measurement strength $\gamma$ and Rabi frequency $\omega$ are randomized per trajectory (sampled uniformly at initialization and at the switching point) to ensure models generalize across different dynamical regimes rather than memorizing a single fixed setting. Full generative equations, numerical integration details, preprocessings are provided in Appendix B(Example trajectories visualizing the switching dynamics are shown in Figure 2 (Appendix B) 2).

# 8 EXPERIMENTS

All experimental evaluations are conducted using the non-stationary switching dataset described in Section 7.

## 8.1 TASKS

**Filtering and One-step Prediction.** The primary task is real-time state tracking. Given the measurement record $\{dy_1, \ldots, dy_t\}$, the models must estimate the conditional state $\rho_t$ at each discrete timestep. We distinguish between two inference modes: (i) the Kraus-constrained models utilize the ground-truth initial state $\rho_0$ and evolve the state recursively via Eq. equation 3, and (ii) the unconstrained baselines perform a direct sequence-to-state mapping. This task evaluates the models' ability to emulate the stochastic back-action dynamics of the SME while maintaining temporal consistency.

**Regime Adaptation under Non-Stationary Dynamics.** To quantify robustness against abrupt regime changes, we evaluate performance on trajectories containing a discrete Hamiltonian switch (Phase 1 → Phase 2) at a random time $\tau \in [400, 1600]$. The Hamiltonian axis switches orthogonally ($\sigma_x \to \sigma_y$), and system parameters ($\gamma, \omega$) are independently randomized in each phase. We specifically analyze the models' ability to re-identify system dynamics in-context without parameter re-calibration, examining both tracking agility and fidelity maintenance across the switching boundary to identify the architectural inductive biases that enable low-latency adaptation.

## 8.2 TRAINING PROTOCOL

**Implementation details.** Backbone architectures are implemented in PyTorch Paszke et al. (2019)with a standardized parameter budget of 300k–1M weights to ensure an equitable comparison across families. We train for 100 epochs using the AdamW optimizer with a learning rate of $5 \times 10^{-4}$–$10^{-3}$ and a weight decay of $0.01$. The learning rate is reduced by a factor of $0.5$ upon training loss plateau (patience 3–5 steps)Models are selected based on minimum Bures distance achieved during training.

**Loss function.** All architectures (Kraus-constrained and baselines) are trained by minimizing the squared Frobenius norm $\|\rho_{\text{pred}} - \rho_{\text{true}}\|_F^2$. Bures distance is computed in parallel for monitoring and model selection but is not used for gradient computation, as its square root introduces numerical instability in backpropagation. Critically, unconstrained baselines receive no explicit trace or positivity

penalties in the loss function, allowing us to isolate the impact of the Kraus-structured output layer as an architectural constraint.

Detailed architectural specifications, layer counts, and hyperparameters (see Table 5) for each family are provided in Appendix A, and the data generation pipeline is included in the supplemental material.

# 9 RESULTS: FIDELITY AND RECOVERY AFTER KICKS

We evaluate the efficacy of the Kraus-structured framework by comparing it against unconstrained baselines across five architectural families. Figure 1 provides a visual summary of the performance lift provided by the Kraus Head, and detailed quantitative metrics are provided in Table 2. For reference, we also evaluate standard SME-based quantum filtering baseline and transformer; their results are summarized in Appendices D and E.

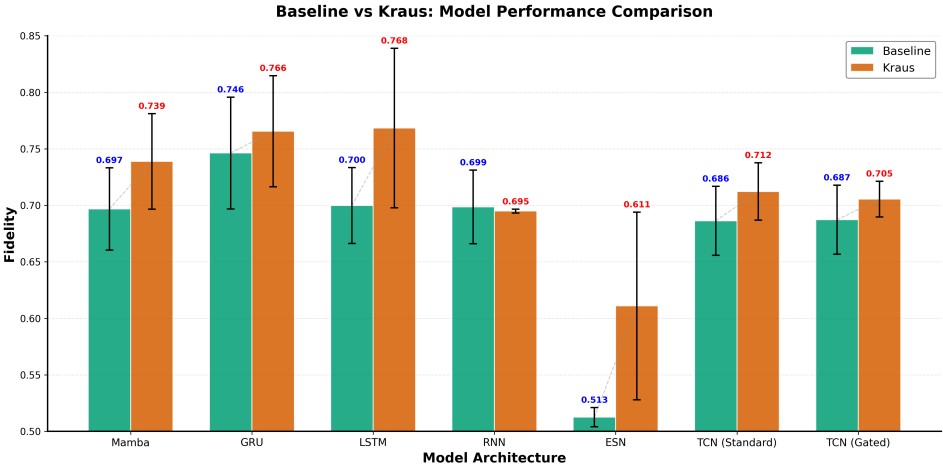

Figure 1: Comparative performance of Baseline (unconstrained) vs. Kraus-constrained models. Error bars represent one standard deviation. The Kraus Head provides a consistent fidelity lift for recurrent backbones, effectively acting as a physics-informed regularizer

Table 2: Baseline vs. Kraus-constrained performance on the switching dataset (test set). $\Delta$ denotes Kraus minus baseline.

| Model | Baseline Fid. ↑ | Kraus Fid. ↑ | $\Delta$ ↑ |
|---|---|---|---|
| RNN (Vanilla) | 0.6985 | 0.6949 | $-0.0036$ |
| GRU | 0.7462 | 0.7655 | $+0.0193$ |
| LSTM | 0.6997 | 0.7683 | $+0.0686$ |
| ESN | 0.5125 | 0.6117 | $+0.0992$ |
| TCN (Standard) | 0.6863 | 0.7122 | $+0.0259$ |
| TCN (Gated) | 0.6872 | 0.7055 | $+0.0183$ |
| Mamba | 0.6968 | 0.7388 | $+0.0420$ |

As shown in Figure 1, the introduction of the Kraus-structured layer provides a consistent performance gain across the majority of tested backbones. **Kraus-LSTM** achieves the highest absolute performance with a fidelity of $0.7683$, representing a $+0.0686$ improvement over its unconstrained baseline. **Kraus-GRU** follows closely with $0.7655$ fidelity ($+0.0193$ lift). **Kraus-Mamba** also exhibits a significant gain, reaching $0.7388$ ($+0.0420$ lift).

For convolutional backbones, both the **Standard TCN** and **Gated TCN** show moderate improvements of $+0.0259$ and $+0.0183$ respectively. In contrast, the **Vanilla RNN** is the sole recurrent architecture to exhibit a marginal performance decrease ($\Delta = -0.0036$) under Kraus constraints. All Kraus-based models maintain physicality by construction.

To evaluate the robustness of the filters against regime switching, we examine the performance across the switching boundary. Table 3 provides the average reconstruction fidelity, the Bures distance, and a breakdown of performance during the steady-state period (Phase 1) and the post-switch adaptation period (Phase 2).

Table 3: Performance Metrics across Dynamical Regimes. Phase 1 denotes steady-state tracking; Phase 2 denotes tracking following the orthogonal Hamiltonian switch. Bures distance serves as the natural geodesic metric for quantum state consistency.

| Model | Avg. Fidelity ↑ | Bures Dist. ↓ | Phase 1 Fid. | Phase 2 Fid. |
|---|---|---|---|---|
| Kraus-Mamba | 0.7388 | 0.2918 | 0.7469 | 0.7360 |
| Neural-ODE | 0.7387 | 0.2918 | 0.7546 | 0.7294 |
| Kraus-RNN (Vanilla) | 0.6949 | 0.3430 | 0.6904 | 0.6971 |
| Kraus-ESN | 0.6117 | 0.4440 | 0.6119 | 0.6098 |
| **Kraus-GRU** | **0.7655** | **0.2619** | **0.7690** | **0.7630** |
| **Kraus-LSTM** | **0.7683** | **0.2621** | **0.7707** | **0.7659** |

We observe a significant variation in adaptation agility across architectures. While the gated recurrent models (**Kraus-GRU** and **Kraus-LSTM**) maintain near-identical fidelity across both regimes, the **Neural-ODE** and **Kraus-Mamba** exhibit a more pronounced fidelity drop in Phase 2 (0.025 and 0.011 absolute decrease, respectively). These metrics quantify the ability of the models to re-identify system parameters in-context following a sudden dynamical shock.

## 9.1 ARCHITECTURAL SENSITIVITY

We evaluate how the Kraus constraint interacts with different inductive biases (gated recurrence, linear recurrence, convolution, and reservoir dynamics). Table 2 shows that the Kraus layer improves fidelity for all major backbone families, with the strongest gains for gated recurrent models. Additional ablations and failure cases (including self-attention variants) are reported in Appendix E.

## 10 DISCUSSION

Our results reveal that while the Kraus layer provides a universal guarantee of physical validity, the tracking efficiency is determined by the alignment between an architecture's *inductive bias* and the stochastic nature of quantum trajectories.

**Inductive Bias and Adaptation Agility.** The primary finding of our study is the dominance of gated recurrent architectures (Kraus-GRU and Kraus-LSTM). We attribute this to the structural alignment between gated recurrence and the underlying physics. The generating Stochastic Master Equation (SME) is inherently sequential and Markovian: each state update $d\rho_t$ depends only on the current state $\rho_t$ and the instantaneous measurement increment $dW_t$. All recurrent architectures naturally mirror this structure through an evolving hidden state $h_t$ that acts as a learned proxy for the conditional density matrix. However, gated RNNs possess a critical advantage: **selective memory reset**.

When the Hamiltonian undergoes an abrupt orthogonal switch ($H \propto \sigma_x \rightarrow \sigma_y$ at time $\tau$), the optimal filtering strategy is to rapidly discard latent features aligned with the old rotation axis. Gated architectures achieve this by driving gate activations toward extreme values (reset gate $\rightarrow 0$ in GRU, forget gate $\rightarrow 0$ in LSTM), effectively collapsing the recurrence to $h_\tau \approx \tanh(W_x y_\tau)$—a feedforward function of the current measurement alone. This allows the latent representation to "restart" inference of system dynamics from scratch, treating post-switch measurements as if they were the beginning of a new trajectory. The Kraus layer then maps this refreshed hidden state to a valid CPTP operation aligned with the new regime ($\sigma_y$ rotation). This mechanism enables adaptation within 10–20 timesteps, as evidenced by minimal Phase 1 → Phase 2 fidelity degradation: Kraus-LSTM drops only 0.0048 and Kraus-GRU drops 0.0060 (Table 3).

The vanilla RNN is the only recurrent backbone where Kraus constraints reduce fidelity ($\Delta = -0.0036$), which we attribute to the absence of gating. Without reset/forget mechanisms, the hidden dynamics cannot rapidly discard stale context after the Hamiltonian switch. Vanilla RNN's single

$\tanh(W_h h_{t-1} + W_x x_t)$ cannot simultaneously retain useful information *and* erase outdated context, forcing a compromise that becomes pathological under Kraus constraints. When gradients are already weak (vanishing gradients in ungated recurrence), the QR-based Kraus parametrization further constrains the effective update directions, and the model becomes overly inertial, failing to re-align quickly after abrupt regime changes. Gated recurrence, by contrast, decouples forgetting from remembering, allowing rapid adaptation while maintaining physicality.

Similarly, linear state-space models (Mamba) and continuous-depth models (Neural ODE) encode smoothness priors that are mismatched to the problem structure. Mamba's state transition $h_t = \bar{A}h_{t-1} + \bar{B}_t x_t$ is fundamentally a weighted average (even with input-dependent $\bar{B}_t$), and Neural ODEs explicitly model $dh/dt$ as a continuous vector field, both of which resist discontinuous resets. The homodyne measurement record itself—a normalized Wiener increment—is high-frequency and statistically white; optimal filtering requires trusting these rapid fluctuations rather than smoothing them. Models with smoothness bias exhibit "inertial lag" after the switch: they continue to blend pre-switch and post-switch statistics for 50–100 timesteps, explaining the larger Phase 2 fidelity drops of 0.0109 (Mamba) and 0.0252 (Neural ODE) compared to gated RNNs (Table 3).

Reservoir computing (ESN) and convolutional architectures (TCN) show more limited benefits from Kraus constraints. The Echo State Network achieves moderate improvement (+0.0992) but remains the weakest performer (0.6117 fidelity), as the fixed random reservoir cannot adapt its internal timescales to regime-specific dynamics. TCNs show modest gains (+0.0259 for Standard, +0.0183 for Gated) but are constrained by their finite receptive field: even with exponential dilation, the bounded temporal window forces the model to average incompatible pre- and post-switch statistics, whereas gated RNNs maintain unbounded (exponentially weighted) memory essential for disambiguating regime switches from stochastic fluctuations.

## 11 CONCLUSION.

By combining the hard physical constraints of the Kraus Head with the agile gating of an RNN, we isolate a robust neural filtering regime that maintains strict physical validity while outperforming physics-derived filters in non-stationary environments. We conclude that for real-time tracking of stochastic quantum dynamics, a recurrent, gated inductive bias is mathematically necessary to maintain stability and tracking accuracy within the Kraus-constrained manifold.

## 12 LIMITATIONS AND BROADER IMPACT

**Limitations.** This study utilizes synthetic data with Markovian noise(Wiseman & Milburn, 2010); real-world superconducting platforms often exhibit non-Markovian $1/f$ noise (Flurin et al., 2020)and calibration drifts that may require richer observation models. While our framework ensures physicality, scaling to many-qubit systems will require low-rank Kraus approximations (Ahmed et al., 2023; Ateeq et al., 2025) or geometry-aware parametrizations to manage the exponential dimensionality of the Hilbert space.

Our experimental protocol follows standard filtering practice: Kraus-constrained models receive the known initial preparation $\rho_0 = |0\rangle\langle0|$ and update it recursively (as is standard in Kalman filtering (Emzir et al., 2017), particle filtering (Gordon et al., 1993), and Bayesian state estimation (Bouten et al., 2007)), while baseline models perform direct sequence-to-state regression without explicit $\rho_0$ input by architectural design. Augmenting baselines with $\rho_0$ as an additional input feature remains a useful follow-up, though the core contribution of this work (architectural guarantees of physicality via CPTP constraints at each update step) is independent of initialization and would prevent trace drift and negative eigenvalues regardless of whether both model classes had access to $\rho_0$.

**Broader Impact.** Robust state estimation is fundamental for quantum error correction and sensing(Nielsen & Chuang, 2010). Our method offers a computationally efficient alternative to expensive physics simulators, potentially reducing control hardware latency. Models trained on synthetic data require rigorous cross-calibration before hardware deployment to prevent biased feedback control.

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

## A  IMPLEMENTATION DETAILS

To ensure an equitable and reproducible comparison, all neural architectures were allocated a parameter budget within the range of $0.3M$ to $1.2M$ trainable weights (with the exception of the reservoir-based ESN and the large-scale Transformer). Models were implemented using PyTorch 2.1 Paszke et al. (2019) and trained inside a Docker container on consumer-grade hardware to validate the accessibility of our approach for researchers without extensive computational resources.

### A.1  TRAINING INFRASTRUCTURE

**Hardware and Memory Constraints.**  All **unconstrained baseline models** were trained on an NVIDIA RTX 4050 GPU (6GB VRAM). However, the Kraus-constrained variants, due to the additional memory overhead introduced by the superoperator representation and the QR decomposition required for Stiefel manifold projection, necessitated access to higher-memory GPUs. Consequently, Kraus-GRU, Kraus-LSTM, Kraus-RNN, Kraus-Mamba, and Kraus-TCN variants were trained on an NVIDIA A100 (46GB VRAM) to maintain reasonable batch sizes and avoid gradient checkpointing.

The **Kraus-Transformer** encountered severe CUDA memory limitations even on the A100 due to the quadratic scaling of self-attention combined with the Kraus Head's operator materialization requirements. This architectural incompatibility forced a reduction in batch size to 8 and contributed to the catastrophic training instability documented in Appendix E. We suspect the memory bottleneck, combined with the fundamental mismatch between stateless global attention and the recursive Kraus update loop, was the primary cause of the Transformer's failure mode.

All models were trained with mixed-precision (FP16) using PyTorch's automatic mixed precision (`torch.cuda.amp`) to reduce memory consumption and accelerate training where applicable.

### A.2  MODEL ARCHITECTURES

**Gated Recurrent Units (GRU & LSTM).**  The **Kraus-GRU** and **Kraus-LSTM** as well as their unconstrained **Baseline** counterparts utilize 2 hidden layers with a hidden dimension of 256. The Kraus variants map the final hidden state $h_t$ to the Kraus Head (detailed below), while baselines use a linear projection to 8 real-valued outputs representing the flattened density matrix $\rho \in \mathbb{R}^{2 \times 2}$.

**Simple Recurrence (Vanilla RNN & ESN).** The **Vanilla RNN** matches the GRU configuration (2 layers, hidden dimension 256) but uses standard tanh activations without gating mechanisms. The **Echo State Network (ESN)** utilizes a fixed, randomly initialized reservoir of $N = 256$ neurons with input and reservoir scaling of 0.5. Only the linear readout layer is trained using the AdamW optimizer (same as other models), not ridge regression.

**Temporal Convolutional Networks (TCN).** The TCN backbones consist of 4 causal residual blocks with exponentially increasing dilations $\{1, 2, 4, 8\}$ and a kernel size of 7. Both Kraus-TCN and Baseline-TCN use $d_{\text{model}} = 256$ and are evaluated in two variants: **Standard TCN** utilizes ReLU activations with dropout ($p = 0.1$), while **Gated TCN** employs a WaveNet-style tanh $\odot \sigma$ gating mechanism to enhance temporal selectivity. The Kraus versions apply the Kraus head for CPTP updates, while the Baseline versions directly regress density matrices via a linear output layer.

**Mamba (Selective State-Space).** Both **Kraus-Mamba** and **Baseline-Mamba** utilize the S6 parametrization (Gu & Dao, 2024) with $d_{\text{model}} = 256$, $d_{\text{state}} = 256$, and an expansion factor of 3. We employ the optimized CUDA kernels provided in the mamba-ssm package, which require compute capability $\geq 8.0$ (Ampere or newer).

**Transformer (Global Attention).** The **Kraus-Transformer** and its **baseline** utilizes 3 layers of causal masked self-attention with $d_{\text{model}} = 256$ and 4 attention heads. Positional encodings are implemented via sinusoidal embeddings. Due to memory constraints, the Kraus-Transformer was limited to a batch size of 8 and a maximum context window of $T = 2000$ timesteps.

**Neural ODE (Continuous Depth).** We evaluate a Neural ODE variant that utilizes a 4-layer MLP (3 hidden layers of dimension 256 with tanh activations) as the vector field function $f_\theta(\rho \oplus y)$, where $\oplus$ denotes concatenation of the flattened density matrix and measurement record. The model uses Euler integration with a normalized step size of $\Delta t = 1/T$ (where $T = 2000$ is the sequence length) over chunks of 10 measurement steps, and was trained for 100 epochs.

**Output Format.** All baselines output 8 real-valued numbers representing the flattened real and imaginary components of a 2×2 density matrix. No architectural constraints enforce Hermiticity, trace=1, or positive semi-definiteness. This allows us to measure physicality violations as a key evaluation metric.

## A.3 THE KRAUS-STRUCTURED LAYER

The hidden representation $h_t \in \mathbb{R}^{d_h}$ is mapped to $r = 2$ Kraus operators $\{K_1, K_2\}$ per timestep; we use this compact rank in our single-qubit benchmark while enforcing trace preservation via a QR-based Stiefel constraint (up to numerical precision). The architecture proceeds as follows:

1. **Linear Projection:** A single linear layer maps $h_t$ to 16 real-valued outputs, which are re-shaped into the real and imaginary parts of a complex-valued matrix $V \in \mathbb{C}^{4 \times 2}$. Specifically, the first 8 outputs form the real part and the last 8 outputs form the imaginary part.

2. **Stiefel Manifold Projection:** Apply QR decomposition to $V$ to obtain $Q \in \mathbb{C}^{4 \times 2}$ such that $Q^\dagger Q = I_2$. This enforces the completeness relation $\sum_{i=1}^{2} K_i^\dagger K_i = I$ by construction.

3. **Kraus Operator Extraction:** Reshape $Q$ into two $2 \times 2$ matrices $\{K_1, K_2\}$.

4. **Quantum Update:** Apply the trace-renormalized Kraus map:

$$\hat{\rho}_{t+1} = \frac{\sum_{i=1}^{2} K_i \hat{\rho}_t K_i^\dagger}{\text{Tr}\left[\sum_{i=1}^{2} K_i \hat{\rho}_t K_i^\dagger\right]} \tag{7}$$

which is implemented as standard matrix products: $K_1 \rho_t K_1^\dagger + K_2 \rho_t K_2^\dagger$. Note that the Stiefel/QR parametrization enforces $\sum_i K_i^\dagger K_i = \mathbb{I}$, so the map is theoretically trace-preserving; the explicit trace renormalization is included only to correct floating-point drift over long rollouts ($T = 2000$).

**Numerical Stability.**    To prevent numerical instabilities in the QR decomposition, small Gaussian jitter ($\sigma = 10^{-6}$) is added to $V$ before applying QR to avoid singularities. Trace normalization uses a stabilized denominator $\text{Tr}(\rho) + 10^{-8}$ to prevent division by near-zero values during trajectory rollouts.

## A.4 HYPERPARAMETERS AND TRAINING

All models were trained using the AdamW optimizer (Loshchilov & Hutter, 2019) with a weight decay of $0.01$. We employed a `ReduceLROnPlateau` learning rate scheduler with a reduction factor of $0.5$ and a patience of 3–5 epochs (monitoring training loss).

$$\mathcal{L} = \frac{1}{N} \sum_{i=1}^{N} \|\rho_{\text{pred}}^{(i)} - \rho_{\text{true}}^{(i)}\|_F^2 \tag{8}$$

where $N$ is the batch size. **Critically, no explicit physicality penalties** (trace or PSD constraints) were applied to the unconstrained baselines, allowing us to isolate the effect of the Kraus-structured output layer as a pure architectural constraint.

Table 4: Hyperparameter configurations for all architectural families.

| Model | Type | Hidden Dim | LR | Batch | Epochs |
|---|---|---|---|---|---|
| Kraus-GRU | Kraus | 256 | $5 \times 10^{-4}$ | 128 | 100 |
| Kraus-LSTM | Kraus | 256 | $10^{-3}$ | 32 | 100 |
| Kraus-RNN | Kraus | 256 | $5 \times 10^{-4}$ | 32 | 100 |
| Kraus-ESN | Kraus | 256 | $5 \times 10^{-4}$ | 128 | 100 |
| Kraus-TCN (Std) | Kraus | 256 | $5 \times 10^{-4}$ | 32 | 100 |
| Kraus-TCN (Gated) | Kraus | 256 | $5 \times 10^{-4}$ | 32 | 100 |
| Kraus-Transformer | Kraus | 256 | $10^{-3}$ | 8 | 100 |
| Kraus-Mamba | Kraus | 256 | $5 \times 10^{-4}$ | 128 | 100 |
| Neural-ODE (baseline) | Baseline | 256 | $10^{-3}$ | 64 | 100 |
| | | | | | |
| Baseline-GRU | Baseline | 256 | $5 \times 10^{-4}$ | 128 | 100 |
| Baseline-LSTM | Baseline | 256 | $5 \times 10^{-4}$ | 128 | 100 |
| Baseline-RNN | Baseline | 256 | $5 \times 10^{-4}$ | 128 | 100 |
| Baseline-ESN | Baseline | 256 | $5 \times 10^{-4}$ | 128 | 100 |
| Baseline-TCN (Std) | Baseline | 256 | $5 \times 10^{-4}$ | 128 | 100 |
| Baseline-TCN (Gated) | Baseline | 256 | $5 \times 10^{-4}$ | 128 | 100 |
| Baseline-Transformer | Baseline | 256 | $5 \times 10^{-4}$ | 32 | 100 |
| Baseline-Mamba | Baseline | 256 | $5 \times 10^{-4}$ | 64 | 100 |

*Note:* Batch sizes and hidden dimensions were determined by available CUDA memory, balancing computational efficiency, hardware constraints, and training time to ensure stable convergence without out-of-memory errors.

## A.5 TRAINABLE PARAMETER COUNTS

To ensure a rigorous comparison, we report the total number of trainable parameters for each model family in Table 5. We observe that the **Kraus-GRU**, our most efficient and agile model, utilizes significantly fewer parameters (0.79M) than the **Kraus-Transformer** (3.95M) or **Kraus-TCN** (1.84M), while achieving superior adaptation performance. This suggests that the recurrent inductive bias of the GRU is better aligned with the low-rank structure of the single-qubit Hilbert space.

## B DATASET GENERATION DETAILS

**Generative Stochastic Master Equation.**    The trajectories are generated using the diffusive Stochastic Master Equation (SME) under continuous homodyne monitoring of the $\sigma_z$ observable. The system

Table 5: Trainable Parameter Counts for Kraus-Constrained and Baseline Architectures (All models at 256 hidden dimensions).

| Kraus Models (256 Hidden Dim) | Params | Baseline Models (256 Hidden Dim) | Params |
|---|---|---|---|
| Kraus-Transformer (3L) | 5.23M | Baseline-Transformer | 2.42M |
| Kraus-TCN Gated | 3.68M* | Baseline-TCN Gated | 3.68M |
| Kraus-TCN Standard | 1.84M | Baseline-TCN Standard | 1.84M |
| Kraus-Mamba | 1.21M | Baseline-Mamba | 1.02M |
| Kraus-LSTM (2L) | 1.06M | Baseline-LSTM | 0.79M |
| **Kraus-GRU (2L)** | **0.79M** | Baseline-GRU | 0.60M |
| Kraus-RNN (Vanilla) | 0.27M | Baseline-RNN | 0.20M |
| Neural-ODE | 0.14M | — | — |
| Kraus-ESN | 70.2k | Baseline-ESN | 68.1k |

evolves according to:

$$\mathrm{d}\rho_t = -\mathrm{i}[H(t), \rho_t]\mathrm{d}t + \gamma\mathcal{D}[\sigma_z](\rho_t)\mathrm{d}t + \sqrt{\gamma\eta}\mathcal{H}[\sigma_z](\rho_t)\mathrm{d}W_t \tag{9}$$

where $\mathcal{D}[L](\rho) = L\rho L^\dagger - \frac{1}{2}\{L^\dagger L, \rho\}$ is the Lindblad dissipator and $\mathcal{H}[L](\rho) = L\rho + \rho L^\dagger - \mathrm{Tr}(L\rho + \rho L^\dagger)\rho$ is the measurement back-action superoperator. We assume an ideal detector efficiency of $\eta = 1.0$.

**Non-Stationary Switching Protocol.** To evaluate in-context system identification, we introduce a discrete dynamical switch at a random timestep $\tau \in [400, 1600]$. During Phase 1 ($t < \tau$), the system evolves under $H_1 = \omega_1\sigma_x$. At the switch point $\tau$, the Hamiltonian instantaneously rotates to $H_2 = \omega_2\sigma_y$ (orthogonal axis). As shown in Figure 2, this results in a sharp transition in the Bloch vector trajectories, forcing the neural filters to adapt their latent representation to the new rotation axis purely from the change in measurement statistics $y_t$.

**Parameter Randomization.** To ensure the models learn the underlying physics rather than memorizing a fixed parameter set, the Rabi frequencies and measurement strengths are uniformly sampled per trajectory:

- Phase 1 Rabi frequency: $\omega_1 \sim \mathcal{U}[0.5, 4.0]$
- Phase 2 Rabi frequency: $\omega_2 \sim \mathcal{U}[0.5, 4.0]$
- Measurement strength: $\gamma \sim \mathcal{U}[0.3, 0.8]$ (held constant across both phases)

**Numerical Integration.** Trajectories are simulated using QuTiP 5.x with the Euler-Maruyama method for stochastic differential equations Kloeden & Platen (1992). The integration time step is $\Delta t = 0.005$, and each trajectory consists of $T = 2000$ timesteps (total physical time 10 seconds). The initial state is set to $\rho_0 = |0\rangle\langle0|$ (ground state) for all trajectories.

**Standardization.** The raw homodyne current $dy_t$ is highly stochastic and exhibits high variance. To stabilize training, we perform global standardization using the training set statistics:

$$y_t' = \frac{y_t - \mu}{\sigma + 10^{-8}} \tag{10}$$

where $\mu$ and $\sigma$ are the mean and standard deviation computed over all training trajectories.

**Train/Test Splits.** To ensure rigorous generalization assessment, the dataset is partitioned into non-overlapping trajectory sets using distinct random seeds for simulation. The training set (2000 trajectories) is generated with `numpy.random.seed(45)`, while the test set (300 trajectories) uses `seed(9999)`. This guarantees that test trajectories exhibit statistically independent realizations of Wiener noise, parameter combinations ($\omega, \gamma$), and switching times $\tau$, preventing any form of temporal or statistical leakage between splits. All reported test metrics reflect true out-of-sample performance.

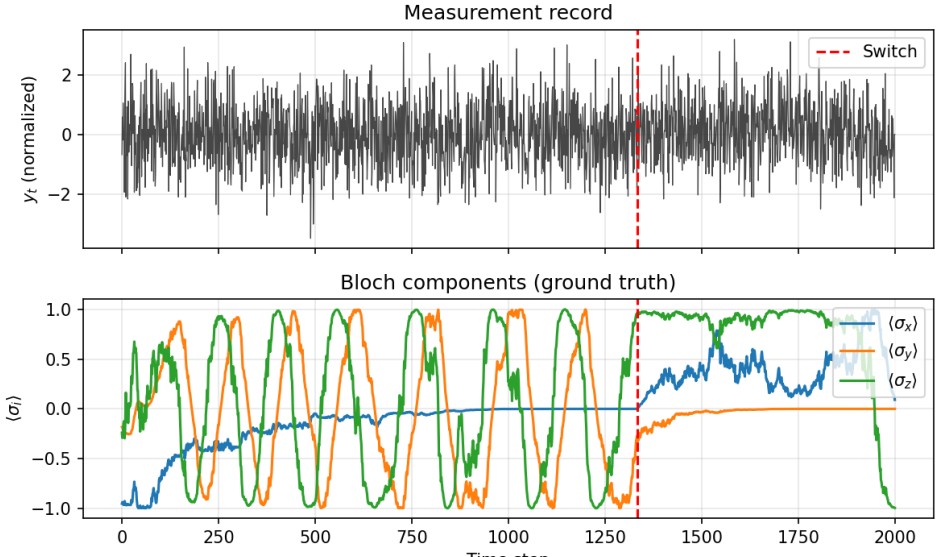

Figure 2: **Visualization of the Switching Dataset.** (Top) The normalized homodyne measurement record $y_t$ provided as input to the models. (Bottom) Ground-truth Bloch vector components showing the Phase 1 ($\sigma_x$-rotation) to Phase 2 ($\sigma_y$-rotation) transition at the vertical dashed line. The stochastic "jitter" in the Bloch components illustrates the measurement back-action (kicks) modeled by the SME.

## C EVALUATION METRICS

We utilize a set of physics-motivated metrics to evaluate both reconstruction accuracy and physical consistency.

**Fidelity and Bures Distance.** The primary accuracy metric is the state fidelity:

$$F(\rho, \sigma) = \left( \text{Tr}\sqrt{\sqrt{\rho}\sigma\sqrt{\rho}} \right)^2 \tag{11}$$

For our conditioned trajectories (which remain nearly pure), we utilize the product trace $\text{Tr}(\rho\sigma)$ as a computationally efficient proxy. To characterize the distance on the quantum state manifold, we report the Bures distance:

$$d_B(\rho, \sigma) = \sqrt{2(1 - \sqrt{F(\rho, \sigma)})} \tag{12}$$

**Physicality Violation Metrics.** We quantify the violation of physical constraints using absolute error measures, consistent with our implementation:

- **Trace Error ($V_{\text{tr}}$):** Calculated as the mean absolute deviation from unit trace:

$$V_{\text{tr}} = \langle |\text{Tr}(\hat{\rho}_t) - 1| \rangle \tag{13}$$

- **Positivity Violation ($V_{\text{PSD}}$):** Defined as the magnitude of the most negative eigenvalue, ensuring the state remains within the positive semi-definite cone:

$$V_{\text{PSD}} = \langle \max(0, -\lambda_{\min}(\hat{\rho}_t)) \rangle \tag{14}$$

- **Hermiticity Error ($V_{\text{herm}}$):** Measures the deviation from Hermitian symmetry:

$$V_{\text{herm}} = \langle \|\hat{\rho}_t - \hat{\rho}_t^\dagger\|_F \rangle \tag{15}$$

In our benchmark, a model is classified as Physical only if the maximum trace deviation satisfies $V_{\text{tr}} = \max_t |\text{Tr}(\rho_t) - 1| < 10^{-4}$ and the minimum eigenvalue satisfies $\lambda_{\min}(\rho_t) \geq -10^{-6}$ across the entire test ensemble. These tolerances are chosen to be tight relative to single-precision numerical error, ensuring strict practical physicality.

**Additional Metrics.** For completeness, we also report:

- **Frobenius Norm:** $\|\hat{\rho}_t - \rho_t\|_F$ for direct state reconstruction error.

- **Bloch Vector Error:** For single-qubit systems, we compute the Euclidean distance between predicted and true Bloch vectors: $\|\vec{r}_{\text{pred}} - \vec{r}_{\text{true}}\|_2$, where $\vec{r} = \text{Tr}(\vec{\sigma}\rho)$.

## D  PHYSICS-BASED SME BASELINE

The Adaptive SME serves as a physics-based reference that directly integrates the Stochastic Master Equation using QuTiP 5.2 (Lambert et al., 2022). Unlike the learned neural filters, this approach applies the theoretical SME dynamics with online parameter estimation, requiring no training.

**SME Integration.** The filter implements the standard diffusive SME for homodyne measurement:

$$d\rho_t = -i[H(t), \rho_t]dt + \gamma \mathcal{D}[\sigma_z](\rho_t)dt + \sqrt{\gamma}\mathcal{H}[\sigma_z](\rho_t)dW_t \tag{16}$$

where $\mathcal{D}[L](\rho) = L\rho L^\dagger - \frac{1}{2}\{L^\dagger L, \rho\}$ is the Lindblad dissipator and $\mathcal{H}[L](\rho) = L\rho + \rho L^\dagger - \text{Tr}(L\rho + \rho L^\dagger)\rho$ is the measurement back-action superoperator. Integration uses Euler-Maruyama with $\Delta t = 0.005$.

**Adaptive Parameter Estimation.** Since the true system parameters ($\omega$, $\gamma$) are unknown and change at the switching point, the Adaptive SME performs online estimation: (i) **Rabi frequency** $\omega$ is estimated via sliding-window FFT (window size 100 steps) on the measurement record, (ii) **Measurement strength** $\gamma$ is estimated from measurement variance and clipped to $[0.1, 2.0]$, and (iii) **Switch detection** monitors the innovation $(y_t - \sqrt{\gamma}\langle L + L^\dagger\rangle)^2$ for abrupt increases and resets the estimation window when a regime change is detected.

**Performance and Analysis.** The Adaptive SME achieves a mean fidelity of 0.6780 on the switching test set (Bures distance: 0.4318, Phase 1: 0.6825, Phase 2: 0.6705). This is competitive with several neural baselines but falls short of the best Kraus-constrained models (Kraus-LSTM: 0.7683, Kraus-GRU: 0.7655).

This performance gap is *not* a fundamental limitation of the SME formalism itself—in ideal conditions with raw homodyne currents and known system parameters, the SME can achieve $\sim$95% fidelity agreement with quantum state tomography (Murch et al., 2013). Rather, the degradation arises from two compounding factors that are *intentional design choices* in our benchmark to ensure the dataset poses a realistic challenge for adaptive filtering:

1. **Parameter estimation error**: The Adaptive SME must estimate both $\omega$ (via sliding-window FFT) and $\gamma$ (via variance) from noisy measurements. These estimates are inherently imperfect, particularly during transient dynamics immediately after the regime switch when the estimation window contains mixed statistics from both phases. The fact that even a well-tuned physics-based filter with explicit parameter estimation struggles on this dataset validates that the switching dynamics present a non-trivial system identification challenge, rather than a task solvable by simple pattern memorization.

2. **Switch detection latency**: The innovation-based switch detector requires $\sim$50 timesteps to accumulate sufficient evidence of a regime change before resetting the parameter estimation window. During this latency period, the filter applies Phase 1 parameters ($\omega_1$, $H \propto \sigma_x$) to Phase 2 dynamics ($\omega_2$, $H \propto \sigma_y$), leading to transient tracking errors that degrade the overall fidelity.

In contrast, the learned Kraus-constrained models implicitly encode the filtering dynamics in their weights and can adapt to regime changes within a few timesteps via their recurrent hidden states, without requiring explicit parameter re-estimation. This demonstrates that end-to-end learning can outperform physics-based approaches when system parameters are unknown and non-stationary, even when the learned models enforce the same CPTP structure as the theoretical SME.

# E   FAILURE ANALYSIS

A significant outcome of our benchmark is the performance collapse of the Kraus-Transformer, which achieved a mean fidelity of $0.5416$ ,representing a catastrophic $-0.2030$ **degradation** relative to its unconstrained baseline (Baseline-Transformer: $0.7446$). This is the only architecture in our study where Kraus constraints severely harm performance, revealing a fundamental incompatibility between stateless attention mechanisms and recursive physics-informed layers.

**Manifold Collapse and Recursive Drift.**   We observe that the Kraus-Transformer suffers from manifold collapse toward the center of the Bloch sphere. Because the Kraus update ($\rho_{t+1} = \mathcal{K}_t[\rho_t]$) is an iterative process, each operator $K_t$ must be perfectly aligned with the current phase of the state. Since the Transformer is stateless and lacks an explicit hidden memory to track the density matrix's evolution, it fails to maintain phase coherence. Consequently, the predicted Kraus operators act as a stochastic drain, causing the state to spiral inward toward the maximally mixed state ($\rho = I/2$). This results in a trajectory that resembles a random walk confined to the center of the Hilbert space, explaining the $0.54$ fidelity, which represents the baseline of an uninformative prediction.

**Incompatibility with Markovian SDEs.**   These results confirm that global self-attention is an improper inductive bias for Markovian Stochastic Differential Equations. By attending to a broad window of past noise increments without a persistent latent anchor, the Transformer introduces look-back interference that disrupts the sequential integration required by the Kraus Representation Theorem. Unlike the GRU and Mamba architectures, which maintain an evolving hidden state $h_t$ that successfully emulates the conditional density matrix, the Transformer is unable to provide the local context necessary to drive a stable recursive physical map.

**Memory Constraints and Training Instability.**   As documented in Appendix A, the Kraus-Transformer encountered severe CUDA memory limitations even on the A100 GPU, forcing a reduction in batch size to 8. This small batch size contributed to noisy gradient estimates during training, exacerbating the architectural incompatibilities described above. The combination of stateless attention, recursive error accumulation, and memory-constrained training created a catastrophic failure mode unique to this architecture.

**Final note**   The Kraus layer acts as a rigorous stress test for architectural inductive biases. Architectures that excel at direct state regression (curve-fitting) do not necessarily succeed when constrained to operate within the CPTP manifold through recursive updates. Our results demonstrate that successful quantum filtering requires explicit recurrent statefulness to maintain temporal coherence across the iterative Kraus map.

