# OpenReview forum: "Kraus Constrained Sequence Learning For Quantum Trajectories from Continuous Measurement"
_ICLR.cc/2026/Workshop/FM4Science — ICLR 2026 Workshop FM4Science Poster_

### Official Review · Reviewer_1Vyn · 2026-02-19
**Kraus Constrained Sequence Learning Review**

**Rating:** 7
**Confidence:** 4

**Review:**

The paper addresses the challenge of real-time quantum state estimation by introducing a Kraus-structured output layer for neural sequence models. This layer ensures that predicted quantum states remain physically valid (completely positive and trace-preserving) by construction, outperforming standard stochastic master equation (SME) solvers in scenarios with parameter drift. The technical execution is robust, featuring a with/without ablation study across several backbone architectures, including RNN, GRU, LSTM, TCN, and Mamba. The paper clearly defines the physical constraints of density matrices (h≥0 and Tr(h)=1) and explains how the Kraus map ensures these are preserved. Adapting Kraus-operator learning from quantum process tomography into a drop-in layer for modern sequence backbones is a significant and novel integration. This work is critical for quantum feedback control, where unphysical estimates from standard neural networks can lead to unstable system rollouts.

Strengths: Ensures physically valid updates by construction, eliminating the need for separate physicality penalties during training. Outperforms traditional physics-derived filters when system parameters are unknown or drifting.The Kraus-LSTM variant improved state estimation quality by 7% over unconstrained models.

Potential areas to address further: While theoretically CPTP, the model still requires numerical stabilizers like Hermiticity projection to cancel floating-point drift over long horizons. Results show that certain architectures like Transformers may struggle with the recursive sequential integration required by the Kraus map.

---

### Official Review · Reviewer_eKjW · 2026-02-20
**The paper proposes a Kraus-structured output layer to enforce physical validity in quantum state estimation by construction. While the Kraus-LSTM shows a 7% fidelity lift over unconstrained models and outperforms physics-based filters , the study relies on simplified and outdated baselines (RNN, GRU, LSTM). It lacks comparison with modern state-of-the-art architectures and fails to investigate if advanced attention variants could resolve the catastrophic failure observed in the Kraus-Transformer.**

**Rating:** 5
**Confidence:** 3

**Review:**

Pros:
Architectural Physicality: The Kraus-structured layer ensures all state updates remain within the CPTP manifold by construction, preventing unphysical estimates like negative eigenvalues.
Empirical Performance: The Kraus-LSTM achieves an improvement in estimation quality over its unconstrained counterpart and outperforms adaptive physics-based filters.
Adaptation Agility: Gated recurrent models demonstrate the ability to re-identify system dynamics within 10-20 timesteps following a sudden regime switch.

Cons:
Outdated and Simplified Baselines: The comparative study relies heavily on relatively older architectures such as RNNs, GRUs, LSTMs (from the 1990s and 2014), and TCNs. While Mamba and Neural ODEs are more modern, the exclusion of state-of-the-art architectures, such as State Space Augmented Transformers or Long-range GNNs, limits the scope of the benchmarking.
Missing Discussion on Modern SOTA: The paper does not address why it omits more advanced, complex models that currently lead sequence modeling leaderboards. There is no discussion regarding the potential performance of these advanced models when constrained by a Kraus head, which is a critical gap given the "foundation model" era of ML research.
Transformer Incompatibility: The catastrophic failure of the Kraus-Transformer (a fidelity drop) suggests that the proposed method may struggle with modern attention-based architectures, yet the paper does not explore whether more advanced Transformer variants could mitigate this.
Scalability and Real-World Noise: The reliance on synthetic Markovian noise and low-dimensional (single-qubit) systems leaves questions about how the Kraus-structured layer would scale to multi-qubit systems or handle non-Markovian noise found in real hardware.

---

### Meta-Review · Area_Chair_1Xpj · 2026-02-27

**Recommendation:** Accept (Poster)
**Confidence:** 3

**Metareview:**

This paper presents a technically sound and conceptually meaningful contribution by embedding quantum physical constraints directly into neural sequence models, ensuring valid density-matrix evolution without auxiliary penalties and demonstrating measurable empirical gains in adaptive estimation tasks. However, the evaluation is limited by unclear Transformer compatibility and constrained experimental scope (synthetic Markovian noise, single-qubit systems). While the integration of Kraus operators as a drop-in physically grounded layer is original and practically relevant for quantum feedback control, there are some concerns about benchmarking breadth and real-world robustness.

---

### Decision · Program_Chairs · 2026-03-03

Accept (Poster)